# Association of Temperament and Character Traits with Suicide Probability, Suicide Attempts, and Perceived Stress Level in Patients with Bipolar Disorder

**DOI:** 10.3390/bs14030197

**Published:** 2024-02-29

**Authors:** Selma Özdemir Yılmaz, Aylin Ertekin Yazıcı, Hamdi Yılmaz

**Affiliations:** 1Department of Psychiatry, Mersin Toros State Hospital, 33060 Mersin, Türkiye; selmazdemir@gmail.com; 2Medical School, Department of Psychiatry, Mersin University, 33079 Mersin, Türkiye; aylin.ertekin@gmail.com; 3Department of Psychiatry, Mersin City Training and Research Hospital, 33240 Mersin, Türkiye

**Keywords:** bipolar disorder, suicide attempt, suicidal behavior, temperament, character, personality, perceived stress

## Abstract

Bipolar disorder (BD) is a high-suicide-risk mental disorder. The purpose of this study was to identify the relationship between temperament and character traits with suicide probability, suicide attempts, and perceived stress level in patients with BD. A total of 39 euthymic patients with bipolar disorder who had a history of suicide attempts and 39 euthymic patients without a history of suicide attempts were included in this study. The sociodemographic and clinical data form, Hamilton Depression Rating Scale (HDRS), Young Mania Rating Scale (YMRS), Structured Clinical Interview for DSM-5-Clinician Version (SCID-5/CV), Temperament and Character Inventory (TCI), Perceived Stress Scale (PSS), and Suicide Probability Scale were used to obtain the data. HDRS, PSS, and SPS scores of the group comprised of patients who attempted suicide were higher than the other group. There was no significant difference between the group of patients who had attempted suicide and the other group in terms of temperament characteristics. In the group of patients who had attempted suicide, self-directedness (SD) and cooperativeness (CO) scores were lower, and the self-transcendence (ST) score was higher than the other group. HA and ST were positively and SD negatively associated with SPS scores. In the regression analysis for suicide risk, the factors most associated with suicide risk were high HDRS and low CO score. Low SD in BD and high ST with CO may be associated with suicide attempts. Alongside low SD, high HA and ST may be associated with suicidal ideation. Treating residual depressive symptoms can reduce the risk of suicide.

## 1. Introduction

Bipolar disorder (BD) is a chronic mental disorder characterized by significant disability and loss of functioning [1,2]. The mortality risk of individuals with BD is increased compared to the general population. These risks include cardiovascular disease, diabetes mellitus, chronic obstructive pulmonary disease, influenza, or pneumonia [3]. Among these risks, suicide is one of them, and it has been reported that the risk is 30 times higher than in the general population, especially without treatment [4]. Between 25 and 50% of individuals with BD have at least one suicide attempt in their past, and 15–20% of these attempts are completed suicides [1,2]. Several risk factors for suicide attempts in these patients were found to be female gender, early onset of the disease, first depressive episode and predominance of depressive episodes, comorbidities of cluster B personality disorder, anxiety disorder, alcohol or substance use disorder, and a history of suicide attempts in first-degree relatives [5].

The relationship between mood disorders and temperament and character has been of interest since ancient times, and the concepts of temperament, character, and personality have been classified in various ways throughout this historical process. Temperament describes genetically and structurally innate characteristics that are not altered throughout an individual’s life. Meanwhile, character broadly refers to learned traits that are acquired via environmental influences and education, that can change over time, and that are learned. On the other hand, personality encompasses the joint contribution of innate temperament and acquired character determinants [6]. Among the people who have contributed the most to the study of temperament and character is Cloninger, who defined personality in terms of two basic components, temperament, and character, and developed a psychobiological model. According to the model, temperament is analyzed in four sub-dimensions: novelty seeking (NS), harm avoidance (HA), reward dependence (RD), and persistence (PS), whereas character is analyzed in three sub-dimensions: self-directedness (SD), cooperativeness (CO) and self-transcendence (ST) [7].

A review of the literature demonstrated that certain temperament and character traits are associated with BD in a situation-independent manner [8]. In previous studies, higher NS and lower PS in temperament subscales and higher ST in character subscale were found in patients diagnosed with BD compared to patients diagnosed with depressive disorder and healthy individuals [9,10,11]; in addition, both higher HA and lower SD were found in mood disorders compared to healthy controls [12]. Pre-illness temperament types were found to have an important role in the clinical development of minor and major mood episodes, including the direction of polarity of acute mood episodes and symptom occurrence. Furthermore, it has been reported that personality traits can significantly influence long-term trajectories and outcomes, including suicidality and other self-mutilative behaviors [10]. Nevertheless, there are different views suggesting that temperament and character traits are a predisposing condition for BD or a manifestation of this disorder [8].

The number of studies investigating temperament and character traits using TCI in patients diagnosed with BD with a history of suicide attempts is limited in the literature [10,13,14,15,16,17]. Engström et al. [15] found high HA and RD in patients diagnosed with BD, who had a history of suicide attempts compared to the group without a history of suicide attempts. High HA, low PS, and low SD were found in patients with BD who had a history of suicide attempts, whereas low RD and low CO were found in patients with BD who did not have a history of suicide attempts compared to healthy controls. They concluded that HA and PS temperament traits are a risk factor for suicide attempts. Sayin et al. [13] and Joyce et al. [14] reported that high HA is a risk factor. According to another study conducted on patients who followed up with a diagnosis of BD, low SD and high ST were found in patients who had attempted suicide [16].

Our study aimed to determine whether the group with a history of suicide attempts and the group without a history of suicide attempts differed in terms of temperament and character traits, to evaluate the perceived stress levels and suicide probabilities, and to determine the factors predicting suicide in euthymic patients with bipolar disorder who were followed up with a diagnosis of BD.

## 2. Materials and Methods

### 2.1. Sample and Procedure

This study included 51 patients with a history of suicide attempts and 54 patients without a history of suicide attempts who were diagnosed with BD as per DSM-5 diagnostic criteria, was followed in remission for at least 3 months, and were admitted to Mersin University Faculty of Medicine Psychiatry Outpatient Clinic between 1 August 2021 and 1 December 2021. Patients included in the study were informed. Written consent was obtained from the patients who volunteered to participate in the study. The approval for this study was obtained from the Mersin University Faculty of Medicine Clinical Research Ethics Committee with the decision dated 28 July 2021 and numbered 2021/512. The inclusion criteria for both groups were determined as being between 18 and 65 years of age, having at least primary school education, not having received electroconvulsive therapy (ECT) in the last 6 months, and having signed the informed consent form.

Following the psychiatric interview according to DSM-5 diagnostic criteria, 5 patients with BD who were determined to be in a depressive state and whose Hamilton Depression Rating Scale score was above 7, 3 patients with comorbid generalized anxiety disorder, and 4 patients who failed to complete the scales were excluded from the study. Among the patients diagnosed with BD without a history of suicide attempts, 3 patients were identified to be in a depressive state and had a Hamilton Depression Rating Scale above 7, 2 patients with hypomanic symptoms and a Young Mania Rating Scale score above 5, 3 patients with comorbid generalized anxiety disorder, 2 patients with comorbid panic disorder, and 5 patients who failed to complete the scales were excluded from the study. Ultimately, the study was conducted with 39 participants from both groups. As a result, all of the patients included in the study were in the remission period and did not have comorbid mental illnesses and substance use disorders. All patients continued medication after remission.

#### Sample Size

The G Power program was employed to calculate the sample size. In order to decide on the sample size, data were collected from 10 patients for both groups and 20 patients in total. The mean and standard deviations of the HA dimension of TCI were calculated. These values were 23.8 ± 6.68 for the group with a suicide attempt; it was found to be 17.7 ± 8.41 for the group without a suicide attempt. With these values, the effect size was calculated as 0.801. The same process was conducted for the Perceived Stress Scale. The values were 22.9 ± 6.98 for the group with suicide attempts; it was found to be 16.7 ± 8.46 for the group without suicide attempts. With these values, the effect size was calculated as 0.799. As a result, the effect size was chosen as 0.8 when determining the sample size. Via the G Power program, the minimum sample size required to reach a statistically significant effect size between the variables was calculated, and it was determined that each group should have 35 participants (effect size d = 0.8; alpha = 0.05; power = 0.95; allocation ration N2/N1 = 1).

### 2.2. Measures

#### 2.2.1. The Sociodemographic and Clinical Data Form

The form included sociodemographic characteristics such as age, gender, marital status, educational status, and disease data related to BD of the patients participating in the study and was filled out by the interviewee.

#### 2.2.2. Hamilton Depression Rating Scale (HDRS)

Developed by Max Hamilton, it is a 17-item scale used to measure the severity of depression, rated by the clinician [18]. The higher scores on the scale indicate an increase in the severity of depressive symptoms. The Turkish validity and reliability study of the scale was conducted by Akdemir et al. and Cronbach’s alpha value was calculated as 0.75 [19].

#### 2.2.3. Young Mania Rating Scale (YMRS)

The scale developed by Young et al. comprises 11 items, each with 5 levels of symptom severity [20]. The coefficient of linear correlation between the total scores of two independent interviewees was 0.93, whereas the coefficients of linear correlation between the scores of each item ranged between 0.66 and 0.92. The reliability and validity study of the scale in Turkish demonstrated that the internal consistency coefficient was 0.79, the consensus among scale items was 63.3–95.5%, and the kappa values were between 0.114 and 0.849 [21].

#### 2.2.4. The Structured Clinical Interview for DSM-5-Clinician Version (SCID-5/CV)

This structured interview consists of 32 diagnostic categories with detailed diagnostic criteria and 17 diagnostic categories, including only the exploratory questions [22]. In the validity and reliability study of the SCID-5/CV Turkish version, the kappa coefficients between the interviewers were found to be between 0.65 and 1.00. The patient himself, the family and relatives, medical documents/records, and the healthcare team were accepted as information sources during the diagnostic interviews [23].

#### 2.2.5. Temperament and Character Inventory (TCI)

It is a self-report scale developed by Cloninger and colleagues, consisting of 240 true/false items [24]. This scale measures 3 character (SD, CO, and ST) and 4 temperament (NS, HA, RD, and PS) dimensions. Turkish validity and reliability studies were carried out, and Cronbach’s alpha values were determined between 0.60 and 0.85 on the temperament scale and 0.82–0.83 on the character scale [25].

#### 2.2.6. Perceived Stress Scale (PSS)

It was developed to determine the extent to which life events are perceived as stressful. It is a 5-point Likert-type self-report scale consisting of 14 questions scored from “never” to “very often”. The 7 items containing positive statements are reverse scored. The higher scores indicate that the stress perception of the individual is high [26]. The Turkish validity and reliability study of the scale was completed, there were 10-item and 4-item forms in addition to the 14-item long form, and the Cronbach’s alpha value of the 10-item form that we used in our study was found to be 0.82 [27].

#### 2.2.7. Suicide Probability Scale (SPS)

It was developed to evaluate suicide risk in adolescents and adults [28]. It is a self-assessment Likert-type rating scale consisting of 36 items. The scale has 4 sub-dimensions: social support/self-perception, anger/impulsivity, hopelessness/loneliness, and suicidal ideation. The adaptation, reliability, and validity study of the IOS for the Turkish population was undertaken, and Cronbach’s alpha coefficient was found to be 0.87 [29].

### 2.3. Statistical Analysis

The data of the study were encoded in the SPSS-21 program for analysis. The data were analyzed in terms of outliers, missing values, and normality. Since the skewness and kurtosis values were within the range of −3 to +3, it was assumed that the data followed a normal distribution. The sociodemographic and disease-related data of the participants were analyzed by descriptive statistical analyses; the chi-square test of independence was used to determine whether there was a difference in the distribution between groups in terms of these data; group differences in terms of scale scores were analyzed by independent groups *t*-test; and the relationships between variables were examined by Pearson correlation and logistic regression analyses. The significance level was accepted as 0.05 for all analyses.

## 3. Results

A total of 39 participants, 24 of whom were female (61.5%) and 15 of whom were male (38.5%), with a history of suicide attempts diagnosed with BD, and 39 individuals, 26 of whom were female (66.7%) and 13 of whom were male (33.3%), without a history of suicide attempts, participated in the study, as shown in Table 1. The average age of the group with a history of suicide attempts was 39.59 (SD = 12.26), while the average age of the other group was 40.02 (SD = 12.33). Considering the distribution in terms of perceived social support, 17.9% of the group with a history of suicide attempts and 43.6% of the participants without a history of suicide attempts reported that the social support they received was sufficient. A statistically significant difference was found between the two groups in terms of social support (*p* < 0.05).

There was a family history of mental illness in 22 patients who had attempted suicide and 14 patients who had not attempted suicide (*p* = 0.069). There was a family history of suicide attempts in 10 patients who had attempted suicide and 5 patients who had not attempted suicide (*p* = 0.151).

A comparison between the scale scores of the group with a history of suicide attempts and the group without a history of suicide attempts is presented in Table 2. Even though our groups were composed of euthymic patients, according to the analysis results, the group with a history of suicide attempts reported higher HDRS (*t* = 5.509; *p* < 0.001), PSS (*t* = 2.623; *p* = 0.011), and SPS (*t* = 2.873; *p* = 0.005) scores than the other group.

In Table 3, when the group with a history of suicide attempts was compared with the group without a history of suicide attempts in terms of temperament and character traits, the group with a history of suicide attempts reported higher ST (*t* = 2.281; *p* = 0.025) scores and lower SD (*t* = −2.860; *p*= 0.005) and CO (*t* = −3.176; *p*= 0.002) scores than the other group. On the other hand, the two groups did not differ in terms of the other scales. The differences between the two groups in terms of temperament and character traits are given in Table 3.

Among the patients who attempted suicide, 23 (59%) had attempted suicide once, 7 (18%) had attempted suicide twice, 3 (7.6%) had attempted suicide thrice, 4 (10.2%) had attempted suicide four times, 1 (2.6%) had attempted suicide five times, and 1 (2.6%) had attempted suicide six times. In the group with a history of suicide attempts, a significant positive correlation was found between the number of suicide attempts and HDRS (*r* = 0.349; *p* = 0.030) and SPS (*r* = 0.589; *p* < 0.001). On the other hand, the number of suicide attempts was negatively correlated with SD scores (*r* = −0.372; *p* = 0.020) and positively correlated with ST (*r* = 0.321; *p* = 0.047). Values related to the results of the analysis are presented in Table 4.

As presented in Table 5, the correlation of temperament and character traits with other variables in the group with a history of suicide attempts revealed that HA scores were correlated with PSS (*r* = 0.426, *p* = 0.007) and SPS (*r* = 0.323, *p* = 0, 045), while there was a strong positive correlation between SD and PSS (*r* = −0.405, *p* = 0.011) and SPS (*r* = −0.509, *p* < 0.001) and a strong positive correlation between ST and SPS (*r* = 0.411, *p* < 0.001).

In the logistic regression analysis performed to examine the relationship between the scores obtained from the measurement tools used in the study and the history of suicide attempts, the group with a history of suicide attempts was coded as “1” and the other group as “0” for the dependent variable suicide attempt. Before the variables were included in the analysis, 50% of the participants were correctly classified. When the classification percentages were analyzed following the inclusion of the variables in the analysis, 76.9% of those without a history of suicide attempts were correctly classified. Among those with a history of suicide attempts, 82.1% were correctly classified. In total, 79.5% of the participants with and without a history of suicide attempts were correctly classified. While this value was 50% at the baseline, an increase was observed as a result of the variables being added to the model. The independent variables account for 50% of the total variance, according to the analysis conducted.

Depression was more associated with the presence of a history of suicide attempts than other variables (B = 0.952; SH = 0.271; *p* < 0.001). Meanwhile, the scores obtained from the CO subscale were associated with the absence of suicide attempts (*B* = −0.223; *SH* = 0.097; *p* = 0.021). Moreover, it was observed that the other scale scores were not related to the presence or absence of a history of suicide attempts. Values related to the results of the analysis are presented in Table 6.

## 4. Discussion

Exploring the complex relationship between temperament and character traits and suicide risk is thought to provide data to be used in suicide prevention, treatment, and the prevention of recurrence. This study aimed to contribute to this field, where suicidal behavior is frequently associated with BD, by using the TCI, which assesses personality dimensionally in a group of patients diagnosed with BD. It is possible to state that the groups of patients diagnosed with BD with and without suicide attempts did not differ in terms of age, gender, marital status, education level, place of residence, and socioeconomic status and that these possible confounding variables were eliminated when comparing these two groups.

In comparative studies with healthy controls, the most frequently repeated finding in terms of temperament characteristics in patients with BD was high HA [6,9,30,31,32,33]. High HA scores are reflected in passive avoidance behaviors such as pessimism, asthenia, fatigue, fear of uncertainty, and being easily fatigued. While the studies have methodological differences, the greatest focus has been on high HA among temperament traits, and particular attention has been paid to its association with a diagnosis of depression [6,31,34]. In a meta-analysis conducted in 2016, the effect of mood on some temperament and character dimensions was consistently confirmed. According to this study, there was a negative correlation between euthymia and HA [35]. In a 23-year follow-up study, high HA was found to be a risk factor for depression [36]. Therefore, it is considered that the effect of depression intensity on HA score is important when analyzing the relationship between high HA score and suicide attempts. The absence of a significant difference between high HA and suicide attempts in our study may be related to the fact that our patient group was euthymic, and both groups were similar in terms of disease-related variables. Nevertheless, it should be stressed that the lack of a healthy control group was also a limitation of the study. Yet, a correlation was found between the increase in HA scores and suicide probability and perceived stress level. A need exists for further studies on the relationship between HA temperament trait and suicidal ideation, suicide attempts, and perceived stress level in this patient group.

No difference was found between the groups in terms of PS temperament traits in our study. It was reported that PS was more common in BD than in healthy controls [16,37,38]. People with low PS temperament are inactive and lazy, give up easily, and are modest and unsuccessful [39]. In a previous study, it was underlined that low PS might have a protective role against suicide [16]. Furthermore, the relationship between low PS and suicide is yet to be clarified. Since there was no control group in our study, it can be said that PS temperament traits did not differ between the groups. Individuals with high PS temperament traits are hardworking, diligent, ambitious, determined, and perfectionist [39]. It has been reported that high PS makes people more susceptible to anxiety disorders rather than mood disorders and may cause positive or negative emotions depending on other personality traits [40].

Literature findings regarding NS and RD temperament characteristics in patients with BD are contradictory. Although high NS scores were found in some studies on BD and/or suicide [35,41,42,43], no significant difference was found in some studies [16,17]. Even low NS scores were found in some studies [9,33]. In our study, no significant difference was detected between such temperament traits and suicide, indicating that prospective studies with large samples are needed to understand the relationship between these temperament traits and suicide.

The character traits frequently observed in patients with BD are low SD [13,16,44,45,46], low CO [13,45,46], and high ST [16,45,46]. Lower SD scores in the group with suicide attempts define such individuals as having difficulty in fighting against difficulties, being weak, fragile, irresponsible, insecure, blaming, unable to establish meaningful internal goals, reactive, avoiding taking responsibility, and having low self-esteem [12,24,25]. It has been reported that low SD is critical for discriminating between patients with mental illness and healthy controls [25]. Engström et al. and Sarısoy et al. found low SD levels in patients diagnosed with BD who had a history of suicide attempts [16,17]. Another study examining personality traits as a mediating factor between suicide and sociodemographic data found low SD [30], and another study examining suicide attempters and healthy controls found low SD and low CO [41]. Moreover, the negative correlation between SD scores and PSS and SPS scores in our study suggests that patients diagnosed with BD with lower SD traits may feel more stressed due to difficulties in problem solving and coping in the face of the difficulties caused by the disease and may feel hopeless easier and may engage in suicidal behavior. Whereas individuals with high SD have high self-esteem, internal focus of control, high problem-solving capacity, and coping behaviors [24]. The available data suggest that the group with high SD character traits may be more resistant to suicide.

Those who score low on the CO are described as self-absorbed, critical, intolerant, vengeful, and opportunistic. Such individuals tend to disrespect other people’s rights and feelings [25,47]. It has been suggested that low SD and CO are indicators of immature defense mechanisms and personality disorder [48,49,50]. Furthermore, Sayın et al. [13] identified low SD and CO in patients with BD accompanying personality disorder. These individuals are more likely to encounter stressors in collectivist societies, and the fact that they do not seek help while struggling with these stressors may be the factors that lead them to suicidal behavior. The significantly lower CO levels detected in the group with suicide attempts in our study are consistent with the studies investigating the character profiles of patients with suicide attempts in the literature [41,42]. Individuals who are dominant in this characteristic have been reported to be empathizing, accepting, supportive, fair, collaborate with others as much as possible, and enjoy helping others [24,51]. According to regression analyses, high scores obtained from this scale were found to be associated with the absence of suicide attempts. Similar to our study, Pawlak et al. found that CO played a protective role in suicide via a regression analysis [52]. In some longitudinal studies conducted on university students, high CO was found to be protective against suicide [53,54,55]. These findings suggest that high CO, particularly when accompanied by high SD, is associated with resilience in coping with specific stressors and may be a protective factor for stress-related mental disorders [53]. Inherently high CO may contribute as a protective factor against suicide by facilitating more adaptive behavior and cooperation.

The individuals with high ST scores, which is the last character trait, are satisfied, patient, and creative. Loftus et al. [46] suggest that high ST accompanying low SD may be an indicator of residual psychotic symptoms. When high ST, low SD, and low CO are combined, the individual may develop magical thinking, rich imagination, and dissociative tendencies that may be accompanied by personality pathologies (especially schizotypal). In contrast, high CO and high SD accompanying high ST are characterized by positive effects such as maturity, spirituality, and creativity [56]. In our study, ST was significantly higher within the group of those who attempted suicide. In addition, we found a correlation between ST scores and SPS scores. The condition causing susceptibility to suicide attempts was thought to be related not only to high ST but also to low SD and low CO. These data were also found in similar studies [16,45,46,57]. Meanwhile, individuals with low ST are impatient, lack imagination and humility, and are prideful. They cannot tolerate uncertainty or surprises. They wish to have control over almost everything. In Western societies, individuals with low ST are often recognized for their rational, scientific, and material achievements. They may have a hard time accepting pain and death [24]. The fact that people with low ST character traits are assertive, individualistic, controlling, and conscious and have difficulty accepting death may be a character trait that protects them against suicide.

In the group composed of individuals who attempted suicide, there was a positive correlation between the number of suicide attempts and HDRS and ST and a negative correlation with SD. In a prospective study by Jylha et al., the number of suicide attempts, both lifetime and prospectively assessed, was found to be associated with high ST, low SD, and CO [32]. The findings are in line with most studies evaluating the association of lifetime suicide attempts with ST, SD, and CO among patients with mood disorders [16,17,52,58]. Among these features, the role of SD was found to be central [32].

Logistic regression analysis to examine the relationship between the scores obtained from the measurement tools used in our study and the history of suicide attempts showed that HDRS scores were more associated with the presence of a history of suicide attempts than other variables. Despite the fact that our sample was composed of euthymic patients, some studies have underlined that residual depressive symptoms have a much more negative impact on functioning than residual manic symptoms [59]. Several prospective studies have reported that the main determinant of suicide was time spent during periods of high-risk illness [60,61,62]. The frequency of suicide attempts was found to increase 25 times in major depressive episodes and 65 times in mixed illness episodes when compared to euthymia [62]. Among patients with mood disorders, suicidal acts without any period of illness have been reported to be rare [60,61,62]. In addition, among the other findings of our study, the fact that the HDRS scores of the group with suicide attempts were higher in the correlation analysis and the HDRS scores increased as the number of suicide attempts increased emphasizes the importance of treating residual depressive symptoms, even though they seem to be subthreshold.

The PSS and SPS scores of the group with a history of suicide attempts were significantly higher than the group without suicide attempts. The distal factors that increase the risk of suicide in BD include genetic factors, personality traits, early childhood trauma, cognitive patterns, and family history of suicide. As for the proximal causes, there are stressful life events, subjective stress perception of the person, current suicidal ideation, and feelings of helplessness and hopelessness. It is possible to state that SPS and PSS can be considered as proximal causes, and when there are predisposing distal causes, they may trigger suicidal behavior [63]. Moreover, a correlation was found between the number of suicide attempts and SPS scores. It was observed that suicide attempts in the past were the most important predictor for suicide risk and that 50% of completed suicide cases had one suicide attempt in the past [64]. This indicates that patients with BD may be a group that is prone to suicidal behavior even in euthymic periods. In a recently published study aimed at differentiating suicide attempt risk profiles in individuals diagnosed with depressive disorder, depressive symptoms, and psychosocial variables (economic difficulties, social isolation, and impairment in leisure time management) were found to be associated with suicide attempts in men [65]. Therefore, it has been reported that interventions should be made to promote interpersonal openness, social support, and behavioral activation against depressive feelings in men. In women, neurocognitive variables, personal independence (professional, financial, and autonomy), and comorbid mental disorders were found to be associated with suicide attempts. Intervention here should focus on supporting the autonomy of female patients with specific training. The focus should be not only on the day-to-day management of external duties but also on the personal perception of psychic efficacy. It has been reported that job counseling and treatment of comorbidities should also be considered among the priorities [65].

Since the sample size in our study was relatively small and the study was cross-sectional, it is difficult to determine the cause-and-effect relationship. BD subtypes are not categorized separately, and studies in larger samples are required to determine the effect of different temperament and character traits of BD subtypes on suicide attempts. Patients included in the study mostly consisted of individuals on regular treatment. BD patients who refused treatment and thus did not receive regular drug treatment or who did not apply to the outpatient clinic for treatment organization were not included in the study. Untreated BD patients may be at higher risk of suicide than those who accept treatment. This situation restricts the generalization of our findings to all BD patients. The method of suicide was not evaluated in patients with suicide attempts. Not including the completed suicide attempts in the study can be considered as another limitation. Finally, since no healthy control group was available, similar and different aspects of temperament and character traits with BD patients could not be evaluated. A need for future studies with a larger sample size, including a control group, and evaluating temperament and character traits in a way to include the method of suicide attempt is apparent.

## 5. Conclusions

Various temperaments and character traits may play an important role in determining suicide attempts. Low SD with low CO accompanying high ST may be associated with suicide attempts in BD. Additionally, residual depressive symptoms may be associated with suicide attempts.

## Figures and Tables

**Table 1 behavsci-14-00197-t001:** Comparison of sociodemographic characteristics of bipolar disorder patients with and without suicide attempts.

	Suicide Attempt (n = 39)		Non-Suicide Attempt (n = 39)			
	N	%	N	%	* χ^2^ *	* p *
Gender						
Female	24	61.5	26	66.7	0.223	0.637
Male	15	38.5	13	33.3		
Marital status						
Single	26	66.7	20	51.3	1.908	0.167
Married	13	33.3	19	48.7		
Education						
Primary school	15	38.5	8	20.5	3.13	0.209
High school	9	23.1	13	33.3		
University	15	38.5	18	46.2		
Living place						
Smaller than province	8	20.5	6	15.4	0.348	0.555
Province	31	79.5	33	84.6		
Occupational status						
Student	7	17.9	2	5.1	4.003	0.135
Employed	12	30.8	18	46.2		
Unemployed	20	51.3	19	48.7		
People with whom they lives						
Alone	6	15.4	1	2.6	6.892	0.075
Mother–father	18	46.2	13	33.3		
Spouse–children	13	33.3	22	56.4		
Other	2	5.1	3	7.7		
Socioeconomic status						
Low	23	59	16	41	2.513	0.113
Middle	16	41	23	59		
Social support						
Sufficient	7	17.9	17	43.6	11.397	0.022 *
Inadequate in some areas	19	48.7	19	48.7		
Inadequate in many areas	11	28.2	2	5.1		
Inadequate in almost every area	1	2.6	1	2.6		
There are none	1	2.6	0	0		
Medical illness						
No	22	56.4	20	51.3	0.206	0.65
Yes	17	43.6	19	48.7		
Smoking						
No	16	41	24	61.5	3.284	0.07
Yes	23	59	15	38.5		

* *p* < 0.05.

**Table 2 behavsci-14-00197-t002:** Comparison of YMRS, HDRS, PSS, and SPS scores of the groups.

	Suicide Attempt (n = 39)	Non-Suicide Attempt (n = 39)	
	Mean ± SD	Mean ± SD	* p *
YMRS	1.20 ± 1.05	0.95 ± 1.00	0.274
HDRS	3.36 ± 1.53	1.43 ± 1.55	<0.001 ***
PSS	21.18 ± 7.58	16.79 ± 7.18	0.011 *
SPS	75.64 ± 14.52	65.38 ± 16.91	0.005 **

* *p* < 0.05, ** *p* < 0.01, *** *p* < 0.001. YMRS: Young Mania Rating Scale, HDRS: Hamilton Depression Rating Scale, PSS: Perceived Stress Scale, SPS: Suicide Probability Scale, SD: standard deviation.

**Table 3 behavsci-14-00197-t003:** Comparison of TCI scores of groups.

	Suicide Attempt (n = 39)	Non-Suicide Attempt (n = 39)	
	Mean ± SD	Mean ± SD	*p*
Novelty seeking	18.69 ± 5.56	17.85 ± 4.55	0.465
Harm avoidance	20.74 ± 7.19	19.20 ± 6.27	0.317
Reward dependence	12.92 ± 3.17	13.77 ± 2.84	0.219
Persistence	4.28 ± 1.83	4.38 ± 1.89	0.808
Self-directedness	23.69 ± 6.24	28.33 ± 7.98	0.005 **
Cooperativeness	26.13 ± 5.63	30.31 ± 5.98	0.002 **
Self-transcendence	19.74 ± 5.20	16.77 ± 6.27	0.025 *

Note: * *p* < 0.05, ** *p* < 0.01.

**Table 4 behavsci-14-00197-t004:** The relationship between the number of suicide attempts and the scales in the group with a history of suicide attempts.

		YMRS	HDRS	NS	HA	RD	PS	SD	CO	ST	PSS	SPS
Number of Suicide Attempts	*r*	0.095	0.349	0.105	0.182	0.029	−0.061	−0.372	−0.224	0.321	0.297	0.589
	*p*	0.566	0.030 *	0.523	0.267	0.861	0.714	0.020 *	0.171	0.047 *	0.067	0.000 ***

* *p* < 0.05, *** *p* < 0.001. YMRS: Young Mania Rating Scale, HDRS: Hamilton Depression Rating Scale NS: novelty seeking, HA: harm avoidance, RD: reward dependence, PS: persistence, SD: self-directedness, CO: cooperativeness, ST: self-transcendence, PSS: Perceived Stress Scale, SPS: Suicide Probability Scale.

**Table 5 behavsci-14-00197-t005:** Relationship between variables in the group with a history of suicide attempts.

		YMRS	HDRS	PSS	SPS
NS	*r*	0.307	0.106	−0.076	0.243
	*p*	0.058	0.521	0.645	0.136
HA	*r*	0.038	0.205	0.426	0.323
	*p*	0.817	0.211	0.007 **	0.045 *
RD	*r*	0.398	−0.189	−0.007	−0.012
	*p*	0.012 *	0.249	0.966	0.942
PS	*r*	−0.221	−0.159	−0.263	−0.186
	*p*	0.177	0.334	0.106	0.258
SD	*r*	−0.166	−0.222	−0.405	−0.509
	*p*	0.313	0.174	0.011 *	0.001 ***
CO	*r*	−0.243	0.019	−0.066	−0.257
	*p*	0.135	0.909	0.690	0.114
ST	*r*	0.187	0.386	0.032	0.411
	*p*	0.254	0.015 *	0.847	0.009 **

* *p* < 0.05, ** *p* < 0.01, *** *p* < 0.001. YMRS: Young Mania Rating Scale, HDRS: Hamilton Depression Rating Scale NS: novelty seeking, HA: harm avoidance, RD: reward dependence, PS: persistence, SD: self-directedness, CO: cooperativeness, ST: self-transcendence, PSS: Perceived Stress Scale, SPS: Suicide Probability Scale.

**Table 6 behavsci-14-00197-t006:** The relationship between suicide attempts and personality dimensions and scale scores in BD patients.

Step		B	Standard Error	Wald	sd	*p*	Exp(B)
1	YMRS	−0.379	0.378	1.005	1	0.316	0.684
	HDRS	0.952	0.271	12.361	1	0.000 ***	2.591
	NS	−0.004	0.070	0.003	1	0.958	0.996
	HA	−0.037	0.058	0.399	1	0.528	0.964
	RD	0.153	0.129	1.403	1	0.236	1.165
	PS	0.017	0.176	0.010	1	0.921	1.018
	SD	0.085	0.071	1.459	1	0.227	1.089
	CO	−0.223	0.097	5.321	1	0.021 *	0.800
	ST	0.092	0.069	1.773	1	0.183	1.097
	PSS	0.092	0.062	2.216	1	0.137	1.097
	SPS	0.068	0.067	1.029	1	0.310	1.070
	Fixed	1.348	3.562	0.143	1	0.705	3.851

* *p* < 0.05, *** *p* < 0.001. YMRS: Young Mania Rating Scale, HDRS: Hamilton Depression Rating Scale NS: novelty seeking, HA: harm avoidance, RD: reward dependence, PS: persistence, SD: self-directedness, CO: cooperativeness, ST: self-transcendence, PSS: Perceived Stress Scale, SPS: Suicide Probability Scale.

## Data Availability

The data presented in this study are available upon request from the corresponding author.

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
