# Peer review of "Association of Temperament and Character Traits with Suicide Probability, Suicide Attempts, and Perceived Stress Level in Patients with Bipolar Disorder"

_behavsci, 2024, doi:10.3390/bs14030197_

Round 1

Reviewer 1 Report

Comments and Suggestions for Authors

Manuscript covers an interesting topic is well designed and well written. I've only minor suggestion for authors:

1) in line 35 authors state: "...among these risks..." but there is no specification abou which are risks other than suicide.

2) in line 70 "MCI" is a refuse for "TCI"

3) for me is not clear if comorbidity with other psychiatric disorders was a exclusion criteria for the study. If so authors should declare it in a more explicit way in material and methods section. If not a table with comorbidity in the sample should be added.

Author Response

1) in line 35 authors state: "...among these risks..." but there is no specification abou which are risks other than suicide.

The following sentence and the reference were added to the 35th line in the text.

''These risks include cardiovascular disease, diabetes mellitus, chronic obstructive pulmonary disease, influenza or pneumonia [3].'' 

reference:

3. Crump, C.; Sundquist, K.; Winkleby, M.A.; Sundquist, J. Comorbidities and mortality in bipolar disorder: a Swedish national cohort study. JAMA Psychiatry2013, 70, 931-939.

2. in line 70 "MCI" is a refuse for "TCI"

MCI was changed to TCI.

''The number of studies investigating temperament and character traits using TCI in patients diagnosed with BD with a history of suicide attempt is limited in the literature [10,13-17].''

3. for me is not clear if comorbidity with other psychiatric disorders was a exclusion criteria for the study. If so authors should declare it in a more explicit way in material and methods section. If not a table with comorbidity in the sample should be added.

In our study, comorbidity with other psychiatric disorders was an exclusion criterion. The following sentence has been added to line 110.

''As a result, all of the patients included in the study were in the remission period and did not have comorbid mental illnesses and substance use disorder.''

Reviewer 2 Report

Comments and Suggestions for Authors

Dear Authors, 

I read with interest your work. Its main goal was to deepen the relationship between temperament, character traits, suicide attempt, suicide probability, and perceived stress in BD patients. 

I do not have major concerns. In my opinion, its main strength is the search for a link between constitutional and behavioral factors within a critical topic, i.e. suicide prevention. The comparison between suicidal and non-suicidal subjects is not easy to find in the literature, therefore your contribution adds more insights in the phenomenon which is usually investigated with single-sample designs. 

The structure of the manuscript is clear, logical and complete. In my opinion, it is referenced adequately. You may consider to add the recent development on suicide prevention given by Network Analysis (PMID: 37922512). I also would like to ask you why you considered an effect size of 0.8 in your power analysis, and to add this explanation in the manuscript. Conclusions are supported by results and are adequately discussed. 

Author Response

1. You may consider to add the recent development on suicide prevention given by Network Analysis (PMID: 37922512).

Network analysis added.

''In a recently published study aimed at differentiating suicide attempt risk profiles in individuals diagnosed with depressive disorder, depressive symptoms and psychosocial variables (economic difficulties, social isolation, and impairment in leisure time management) were found to be associated with suicide attempt in men [65]. Therefore, it has been reported that interventions should be made to promote interpersonal openness, social support and behavioral activation against depressive feelings in men. In women, neurocognitive variables, personal independence (professional, financial and autonomy) and comorbid mental disorders were found to be associated with suicide attempt. Intervention here should focus on supporting the autonomy of female patients with specific training. The focus should be not only on the day-to-day management of external duties, but also on the personal perception of psychic efficiacy. It has been reported that job counseling and treatment of comorbidities should also be considered among the priorities [65].''

reference: 

Sarti, P.; Colliva, C.; Varrasi, S.; Guerrera, C.S.; Platania, G.A.; Boccaccio, F.M.; Castellano, S.; Pirrone, C.; Pani, L.; Tascedda, F.; et al. A network study to differentiate suicide attempt risk profiles in male and female patients with major depressive disorder. Psychol. Psychother. 2024, 31, 2924.

 2. I also would like to ask you why you considered an effect size of 0.8 in your power analysis, and to add this explanation in the manuscript. Conclusions are supported by results and are adequately discussed. 

''In order to decide on the sample size, data were collected from 10 patients for both groups and 20 patients in total. The mean and standard deviations of the HA dimension of TCI were calculated. These values were 23.8 +- 6.68 for the group with a suicide attempt; It was found to be 17.7 +- 8.41 for the group without suicide attempt. With these values, the effect size was calculated as 0.801. The same process was conducted for the perceived stress scale. 22.9+- 6.98 for the group with suicide attempts; It was found to be 16.7 +- 8.46 for the group without suicide attempt. With these values, the effect size was calculated as 0.799. As a result, the effect size was chosen as 0.8 when determining the sample size.'' 

This section was added to the sample size.

Reviewer 3 Report

Comments and Suggestions for Authors

Bipolar disorder is a widely discussed topic in academic papers, with most of them focusing on diagnosis and treatment. However, you have addressed a critical aspect, making your research thoughtful and interesting.

Suggestions and corrections

Social Support Definition- How did the authors create the sub-sections?

Inclusion: Patients with Bipolar Disorder, continued medication after remission? Please clarify that?

What about family history of BD and suicide, substance use? – Was it accounted for inclusion/exclusion criteria

In our study, ST was significantly higher within the group of those who attempted suicide (This was interesting, but could this be also due to more females?)

My impressions: Interesting thought, but the population size was limiting, 

Comments on the Quality of English Language

MINOR CORRECTIONS 

Author Response

1. Social Support Definition- How did the authors create the sub-sections?

Subsections were created by us in a semi-structured manner. Family support, support from someone who means something special to the individual, and support from friends were questioned. If there are no deficiencies, it is "sufficient"; if there are deficiencies in one, it is "inadequate in some areas"; if there are deficiencies in two, it is " inadequate in many areas"; if there are deficiencies in three, it is " inadequate in almost every area"; if it reports that it has no social support, '’there are none’' option has been selected.

2. Inclusion: Patients with Bipolar Disorder, continued medication after remission? Please clarify that?

''All patients continued medication after remission.'' This sentence was added to the sample and procedure section in the text.

 3. What about family history of BD and suicide, substance use? – Was it accounted for inclusion/exclusion criteria

''There was a family history of mental illness in 22 patients who attempted suicide and 14 patients who did not attempt suicide (p=0.069). There was a family history of suicide attempt in 10 patients who attempted suicide and 5 patients who did not attempt suicide (p=0.151).'' This sentence was added to the results section in the text.

''As a result, all of the patients included in the study were in the remission period and did not have comorbid mental illnesses and substance use disorder.'' This sentence was added to the sample and procedure section in the text.

4. In our study, ST was significantly higher within the group of those who attempted suicide (This was interesting, but could this be also due to more females?) 

In our study, the number of women was higher in both groups. While it was 24 in the group with a suicide attempt, it was 26 in the group without a suicide attempt. Therefore, the number of women was higher in the group without suicide attempts. It has been reported in the literature that women score particularly high on HA and RD (PMID:17292707). However, we did not measure groups by gender in our study.

5. My impressions: Interesting thought, but the population size was limiting, 

The limited number of samples was added as a limitation of the study.

''Since the sample size in our study was relatively small and the study was cross-sectional, it is difficult to determine the cause-and-effect relationship. ''